# Computational modeling of brainstem circuits controlling locomotor frequency and gait

Jessica Ausborn[1†]*, Natalia A Shevtsova[1†], Vittorio Caggiano[2], Simon M Danner[1], Ilya A Rybak[1]

[1]Department of Neurobiology and Anatomy, College of Medicine, Drexel University, Philadelphia, United States; [2]IBM TJ Watson Research Center, Yorktown Heights, United States

**Abstract** A series of recent studies identified key structures in the mesencephalic locomotor region and the caudal brainstem of mice involved in the initiation and control of slow (exploratory) and fast (escape-type) locomotion and gait. However, the interactions of these brainstem centers with each other and with the spinal locomotor circuits are poorly understood. Previously we suggested that commissural and long propriospinal interneurons are the main targets for brainstem inputs adjusting gait (Danner et al., 2017). Here, by extending our previous model, we propose a connectome of the brainstem-spinal circuitry and suggest a mechanistic explanation of the operation of brainstem structures and their roles in controlling speed and gait. We suggest that brainstem control of locomotion is mediated by two pathways, one controlling locomotor speed via connections to rhythm generating circuits in the spinal cord and the other providing gait control by targeting commissural and long propriospinal interneurons.

DOI: https://doi.org/10.7554/eLife.43587.001

*For correspondence:
ja696@drexel.edu

†These authors contributed equally to this work

**Competing interests:** The authors declare that no competing interests exist.

## Introduction

To survive in changing and unpredictable environments animals need to continuously adapt their behavior including locomotor speed. In quadrupeds, changes in locomotor speed are accompanied by changes in interlimb coordination (*Grillner, 1975*; *Miller et al., 1975*; *Hildebrand, 1989*; *Maes and Abourachid, 2013*). During relatively slow locomotion, for example when animals explore the environment, they typically exhibit left-right asynchronous/alternating gaits, such as walk and trot. Alternatively, during chasing/hunting or escaping threats that require faster movements, animals switch to left-right synchronous gaits, such as gallop and bound (*Clarke and Still, 1999*; *Herbin et al., 2004*; *Herbin et al., 2007*; *Bellardita and Kiehn, 2015*; *Lemieux et al., 2016*). Although the generation of locomotor oscillations and mutual interactions between oscillators controlling each limb are implemented within the spinal cord (*Grillner, 2006*; *Kiehn, 2006*; *Goulding, 2009*; *Grillner and Jessell, 2009*; *Kiehn, 2016*; *Boije and Kullander, 2018*), both locomotor speed and interlimb coordination (gait) are controlled by several brainstem structures that transform signals from higher brain centers into meaningful commands to initiate, stop or modulate locomotor frequency and gait (*Lemon, 2008*; *Ryczko and Dubuc, 2013*; *Roseberry et al., 2016*; *Kim et al., 2017*; *Takakusaki, 2017*; *Brownstone and Chopek, 2018*; *Ferreira-Pinto et al., 2018*; *Gatto and Goulding, 2018*).

The advent of viral and genetic tools recently enabled experimental studies to further dissect the brainstem locomotor command circuitry into functionally distinct neuronal populations. A series of studies have identified such populations, their projections, interactions and downstream targets in the mesencephalic locomotor region (MLR) and the pontomedullary reticular formation (RF) that are

critically involved in the control of locomotor speed and gait (*Capelli et al., 2017*; *Caggiano et al., 2018*; *Josset et al., 2018*). The MLR is composed of two main nuclei: the cuneiform nucleus (CnF) and the pedunculopontine nucleus (PPN). Glutamatergic neurons in both nuclei contribute to slow alternating-gait locomotion, while only glutamatergic neurons in the CnF are involved in eliciting fast synchronous-gait locomotion (*Caggiano et al., 2018*; *Josset et al., 2018*). Glutamatergic neurons from both nuclei project to the RF, including the lateral paragigantocellular nucleus (LPGi), and transsynaptically activate the spinal locomotor circuits via the reticulospinal tract (*Capelli et al., 2017*; *Brownstone and Chopek, 2018*). Stimulation of glutamatergic neurons in the LPGi with increasing intensities results in progressively higher locomotor speeds (*Capelli et al., 2017*; *Oueghlani et al., 2018*), which are typically associated with bound/gallop gaits (*Bellardita and Kiehn, 2015*). In addition, optogenetic activation of inhibitory neurons in the MLR (*Roseberry et al., 2016*) or selectively in the CnF or PPN (*Caggiano et al., 2018*) reduces locomotor speed and - depending on the targeted nucleus and initial locomotor speed - can halt locomotion completely. Selective activation of inhibitory neurons in the RF, for example in the gigantocellular nucleus (Gi) or the LPGi, also slow down and can even stop locomotion (*Capelli et al., 2017*). While great progress has been made in identifying and probing these brainstem areas, the exact pathways and circuit organization by which they interface with the spinal locomotor circuitry to control locomotor activity are still unclear.

We previously proposed computational models of the spinal locomotor circuits controlling inter-limb coordination and speed dependent gait expression in intact mice and in mutants lacking specific types of genetically identified commissural interneurons (CINs) or long propriospinal neurons (LPNs; *Danner et al., 2016*; *Danner et al., 2017*). Based on the modeling results, we suggested that CINs and LPNs are the main targets for supraspinal (and other, e.g. afferent) inputs adjusting gait. Our previous models describe the spinal locomotor circuitry and operate under control of external 'brainstem' drives without considering their specific origin and pathways. The new experimental data on the brainstem centers controlling locomotion (*Capelli et al., 2017*; *Caggiano et al., 2018*; *Josset et al., 2018*), allowed us now to extend our previous models by including the brainstem locomotor centers and simulating their possible interactions with the spinal locomotor circuitry. Specifically, we extended the model of *Danner et al. (2017)*, which consisted of four rhythm generators (RGs, each controlling one limb) interacting via CINs and LPNs, to incorporate the bilaterally interacting CnF and PPN circuits and their LPGi-mediated descending pathways to the spinal cord. The suggested organization of synaptic inputs from these pathways to the spinal RGs, CINs and LPNs allowed the model to reproduce the experimentally observed effects of stimulation of excitatory and inhibitory neurons within the CnF, PPN, and LPGi.

Using the model, we investigated (a) the involvement of CnF and PPN in the control of low-frequency alternating-gait locomotion, (b) the specific role of the CnF in the control of high-frequency synchronous-gait locomotion, and (c) the roles of inhibitory neurons located in these brainstem areas in modulating and/or stopping locomotion. Specifically, our simulations have shown that the suppression of glutamatergic PPN neurons during CnF stimulation-evoked locomotion can lead to a shift of the transition from trot to gallop/bound towards lower locomotor frequencies. We suggest that brainstem control of locomotion is mediated by two pathways, one controlling frequency and speed via connections to the rhythm generating circuits and the other controlling gait expression via connections to CINs and LPNs.

## Results

### Model description

To model and computationally investigate the brainstem control of locomotion, we built upon our previous model of spinal circuits, consisting of four RGs (each controlling one limb), which interact via local cervical and lumbar CINs and LPNs connecting cervical and lumbar compartments (*Danner et al., 2017*).

In our previous model, the control of locomotor frequency and gaits was provided by changes of tonic excitatory inputs to the RGs and inhibitory inputs to the particular spinal CIN and LPN

populations. These inputs were considered as external brainstem drives. The implemented network architecture allowed the model to reproduce multiple experimental data, including a monotonic increase of locomotor frequency and frequency-dependent sequential gait transitions from walk to trot and then to gallop and bound (*Shik et al., 1966*; *Shik and Orlovsky, 1976*; *Bellardita and Kiehn, 2015*), when external drives progressively increased (*Danner et al., 2017*). Moreover, the model was able to reproduce changes in frequency-dependent gait expression in mutant mice lacking specific genetically identified CINs and LPNs (*Bellardita and Kiehn, 2015*). Here we have extended this model by incorporating brainstem compartments that include bilaterally located MLR and RF structures providing descending brainstem drives to the spinal cord.

Figure 1 shows a simplified schematic of the extended model and illustrates our main assumptions concerning circuit organization in the brainstem providing inputs to spinal circuits controlling locomotion. The full schematic of the model is shown in *Figure 2*. The MLR on each side includes CnF and PPN structures and the RF is represented by the LPGi. The CnF, PPN and LPGi include

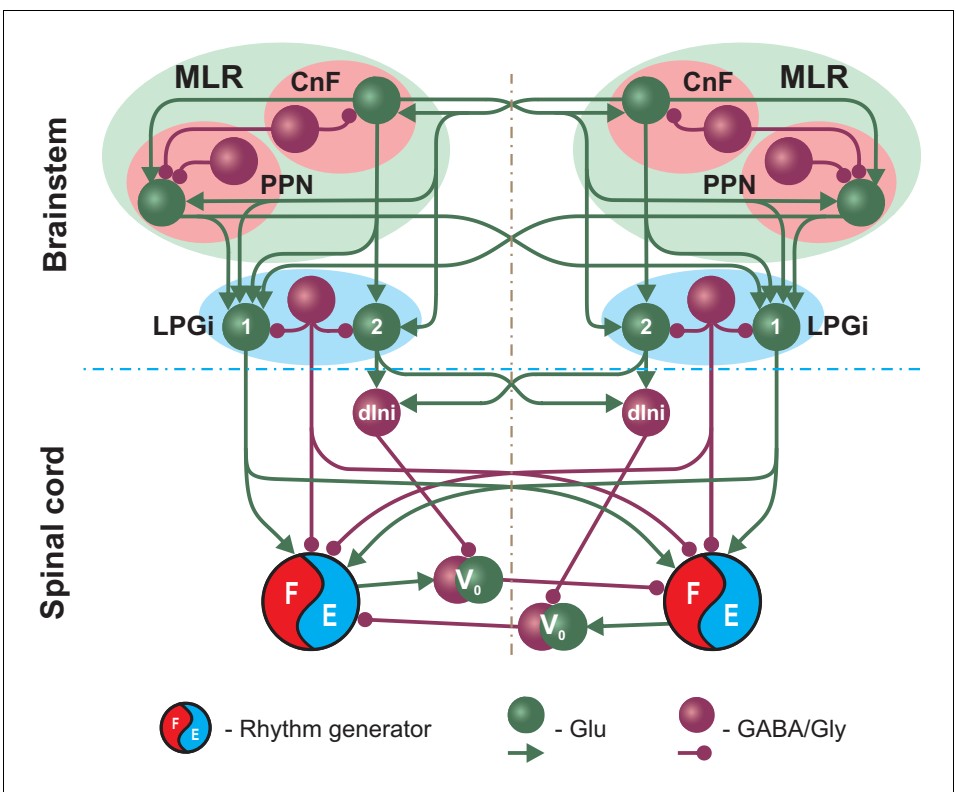

**Figure 1.** Simplified schematic illustrating the model concept for the brainstem control of locomotion. The brainstem compartment on each (left and right) side contains three major structures: the cuneiform (CnF) and pedunculopontine (PPN) nuclei, comprising the mesencephalic locomotor region (MLR), and the pontomedullary reticular formation represented by the lateral paragigantocellular nucleus (LPGi). Each of these structures contain excitatory (glutamatergic, Glu) and inhibitory (Gly/GABA) populations. The bilaterally interacting CnF and PPN control spinal circuits, including rhythm generators (RGs), by descending drives originating from their glutamatergic populations and mediated by the bilaterally located LPGi. Spinal projections from each LPGi are organized in two pathways involving two distinct glutamatergic LPGi populations: '1' and '2'. The LPGi-Glu-1 population relays excitation from glutamatergic neurons in both CnF and PPN and projects to the rhythm generating circuits in the spinal cord. This pathway controls locomotor frequency. The LPGi-Glu-2 population relays excitation from the CnF and projects to inhibitory relay neurons (dIni) in the spinal cord controlling the activity of V0 commissural neurons securing left-right interactions between the RGs and therefore locomotor gait. For simplicity, only the left-right RGs and their connections for the cervical spinal cord are shown. Spheres represent neuronal populations and lines represent synaptic connections with arrowheads for excitatory and circles for inhibitory influences.
DOI: https://doi.org/10.7554/eLife.43587.002

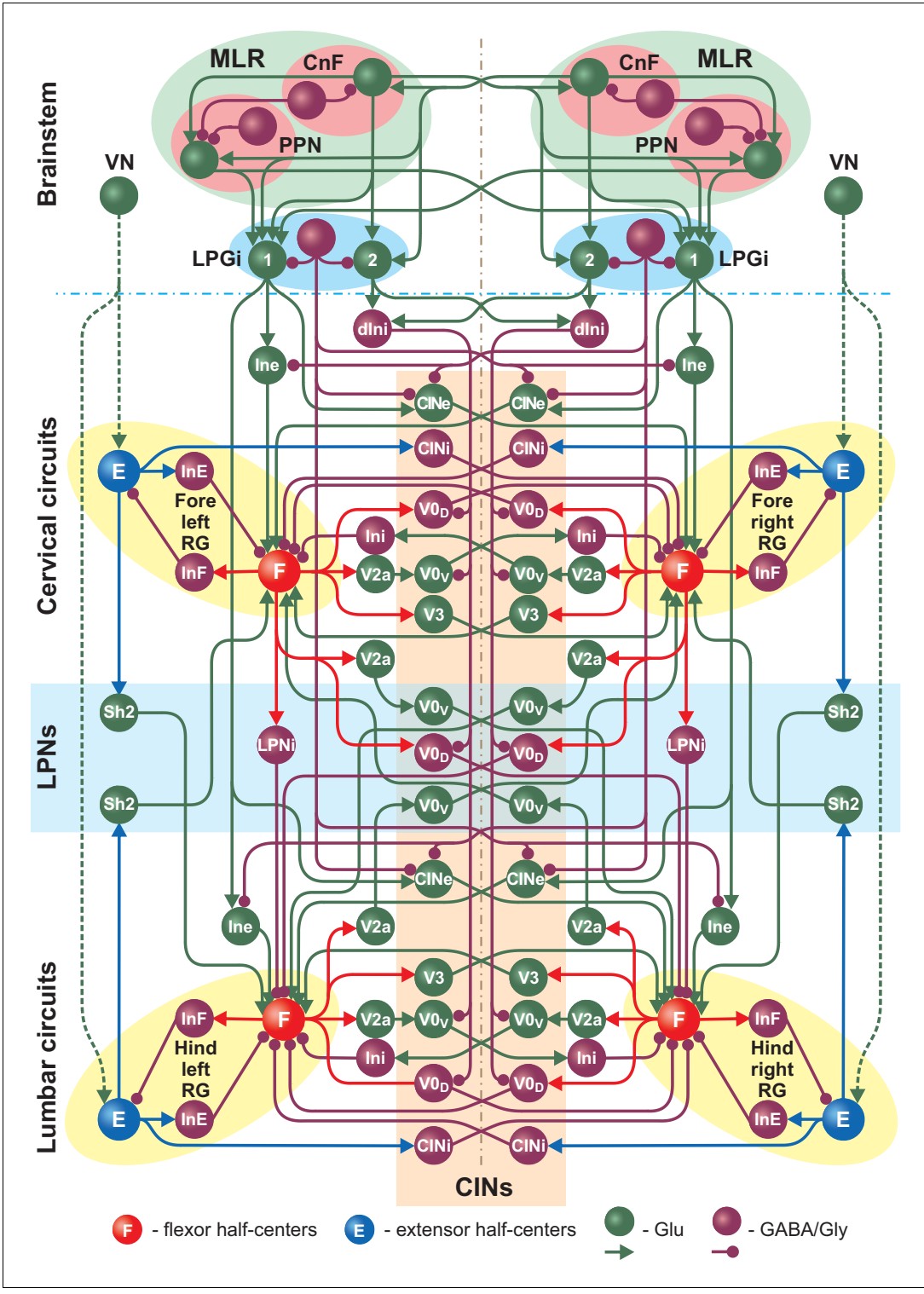

**Figure 2.** Full model schematic showing interactions between the brainstem and spinal cord (cervical and lumbar) circuits. The structure of the cervical and lumbar circuits and their connections is taken from *Danner et al. (2017)*. Brainstem circuits include the PPN and CnF compartments in the MLR and the LPGi compartment in the reticular formation. The LPGi project to the spinal cord via a set of interneuronal populations (see Results). Spheres represent neuronal populations and lines represent synaptic connections with arrowheads for excitatory and circles for inhibitory influences.

DOI: https://doi.org/10.7554/eLife.43587.003

excitatory glutamatergic (Glu) and inhibitory GABAergic or glycinergic (GABA/Gly) neurons. Synaptic interactions within and between the CnF, PPN and LPGi on each side and their bilateral connections are organized according to the existing direct and indirect experimental data. Glutamatergic neurons in the CnF project to the contralateral CnF and to the ipsi- and contralateral PPN and LPGi (*Figures 1* and *2*; *Caggiano et al., 2018*). Glutamatergic neurons in the PPN project to the ipsi- and contralateral LPGi neurons (*Figures 1* and *2*; *Caggiano et al., 2018*). Inhibitory neurons in the CnF and PPN have local projections (*Caggiano et al., 2018*). LPGi inhibitory neurons project locally as well as to the ipsi- and contralateral spinal circuits in the cervical and lumbar compartments (*Figure 2*; *Capelli et al., 2017*).

In the present model, we hypothesized the existence of two distinct populations of glutamatergic neurons in each LPGi (LPGi-Glu-1 and LPGi-Glu-2, labeled in *Figures 1* and *2* as '1' and '2', respectively, and, correspondingly, two distinct pathways from these populations to the spinal cord circuits. The LPGi-Glu-1 populations receive bilateral excitation from the glutamatergic neurons of both CnFs and PPNs and project to the flexor centers of the homolateral (via excitatory interneuron populations Ine) and the contralateral (via the descending commissural populations CINe) RGs in both cervical and lumbar compartments (*Figure 2*). This allows these LPGi populations and their descending pathways to control the frequency of locomotor oscillations generated by the RGs.

In contrast, the LPGi-Glu-2 populations receive bilateral excitatory inputs only from the CnFs and project to descending spinal interneurons (dIni) inhibiting the cervical and lumbar $V0_V$ and $V0_D$ CINs and the descending $V0_D$ LPNs (*Figure 2*). This allows these LPGi populations and corresponding pathways to influence limb coordination and gait.

To keep the extensor centers in a tonic mode (*Zhong et al., 2012*; *Shevtsova et al., 2015*; *Danner et al., 2016*; *Shevtsova and Rybak, 2016*; *Danner et al., 2017*; *Ausborn et al., 2018*) a constant input drive ($D_{VN,ex}$ = 2.15) was applied to the left and right brainstem VN populations which project to all four RG extensor centers (E, *Figure 2*, see also *Danner et al., 2016*). The VN populations in our model represent vestibular nuclei as well as other potential sources of excitatory inputs to the extensor centers of the central pattern generator involved in postural control. Vestibulospinal neurons are known to be tonically active (*Orlovsky, 1972*), preferentially project to spinal interneurons and slow motoneurons of the extensor pools (*Miller et al., 1975*; *Lemon, 2008*; *Basaldella et al., 2015*) and have been implicated in mediating extensor tone (*Fulton et al., 1930*; *Burke et al., 1972*). However, other supraspinal centers might also be involved. All other lumbar and cervical circuits and their local, ascending and descending interconnections shown in *Figure 2* are described in *Danner et al. (2017)*. Connection weights are listed in *Table 1*.

## Differential role of multiple brainstem centers

In their recent studies, *Caggiano et al. (2018)* and *Josset et al. (2018)* have explored the anatomical and molecular heterogeneity of the MLR, highlighting the differential role of glutamatergic neurons within the CnF and PPN. Our model was implemented and adjusted to reproduce their main results:

1. Unilateral selective activation of glutamatergic CnF neurons in the mouse can drive the full range of locomotor speeds with speed-dependent gait expression, including walk, trot, gallop and bound (*Caggiano et al., 2018*; *Josset et al., 2018*). In our model, progressively increasing the excitatory tonic drive ($\alpha$, see Materials and methods) to the glutamatergic population of the left CnF resulted in an increase in locomotor frequency (upper panel in *Figure 3A*) up to 10–11 Hz. This increase of frequency occurred because the applied stimulation produced a progressive activation of the glutamatergic population in the contralateral CnF and in both PPNs, and finally a progressive activation of both LPGi-Glu-1 populations. The LPGi-Glu-1 populations, via the intermediate left and right Ine and CINe populations, progressively activated the flexor centers (F) of all four RGs (*Figure 2*) causing the observed monotonic increase of locomotor frequency. Simultaneously, stimulation of the glutamatergic population of the left CnF resulted in a progressive activation of both LPGi-Glu-2 populations, which, via both descending inhibitory dIni, progressively inhibited cervical and lumbar $V0_V$ and $V0_D$ CINs and descending $V0_D$ LPNs, leading to a stimulation-dependent sequential gait transition from walk to trot and then to gallop and bound. This is illustrated in *Figure 3A* where left-right, homolateral and diagonal phase differences between the four RGs characterize the different gaits. *Figure 3—figure supplement 1* shows examples of extensor activity and polar diagrams for each gait. For definitions and a detailed explanation of gait characteristics see *Danner et al.*

**Table 1.** Connection weights.

| Source | Target ($w_{ij}$) |
| --- | --- |
| **Within brainstem** | |
| CnF-Glu | i-PPN-Glu (0.56), i-LPGi-Glu-1 (0.95), i-LPGi-Glu-2 (1.02), c-CnF-Glu (0.1), c-PPN-Glu (0.15), c-LPGi-Glu-1 (0.45),c-LPGi-Glu-2 (0.08), |
| CnF-GABA/GLY | i-CnF-Glu (−0.5), i-PPN-Glu (−0.5) |
| PPN-Glu | i-LPGi-Glu-1 (1), c-LPGi-Glu-1 (0.4) |
| PPN-GABA/GLY | i-PPN-Glu (−0.5), |
| LPGi-GABA/GLY | i-LPGi-Glu-1 (−0.5), i-LPGi-Glu-2 (−0.5) |
| **Vestibular input to spinal cord** | |
| VN | i-RG-E (1) |
| **From brainstem to relay neurons** | |
| LPGi-Glu-1 | i-Ine (1), i-CINe (1) |
| LPGi-Glu-2 | i-dIni (1), c-dIni (1) |
| LPGi-GABA/GLY | c-Ine(−0.5), c-CINe (−0.5) |
| **From relay neurons to spinal circuits** | |
| Ine | i-RG-F (1) |
| Ini | i-V0$_D$ (4), i-V0$_V$ (1.7), i-f-V0$_D$-LPN (7.5) |
| CINe | c-RG-F (1) |
| **Within girdle and side of the cord** | |
| RG-F (fore and hind) | i-InF (0.4), i-V0$_D$ (0.7), i-V2a-lr (1), i-V3 (0.35), i-V2a-diag (0.5) |
| f-RG-F (fore only) | i-LPNi (0.7), i-V0$_D$-LPN (0.5) |
| RG-E | i-InE (0.4), i-CINi (0.4), i-Sh2-LPN (0.5) |
| InF | i-RG-E (–1) |
| InE | i-RG-F (–0.08) |
| V2a-lr | i-V0$_V$ (1) |
| V2a-diag | i-V0$_V$-LPN (0.9) |
| Ini | i-RG-F (–0.075) |
| **Between left and right circuits within a girdle** | |
| V0$_D$ | c-RG-F (–0.07) |
| V0$_V$ | c-Ini (0.6) |
| V3 | c-RG-F (0.03) |
| CINi | c-RG-F (–0.03) |
| **Between fore and hind circuits** | |
| f-LPNi | ih-RG-F (–0.01) |
| f-Sh2-LPN | ih-RG-F (0.01) |
| h-Sh2-LPN | if-RG-F (0.075) |
| f-V0$_D$-LPN | ch-RG-F (–0.1) |
| f-V0$_V$-LPN | ch-RG-F (0.02) |
| h-V0$_V$-LPN | cf-RG-F (0.065) |

i-, ipsilateral; c-, contralateral; f-, fore; h-, hind; CINi, inhibitory commissural interneurons; Ini, inhibitory interneurons; InE, extensor center inhibitory interneuron; InF, flexor center inhibitory interneuron; LPNi, inhibitory long propriospinal neuron; dIni, inhibitory relay neurons; Ine, ipsilaterally projecting tonically active excitatory relay neurons; CINe, commissural tonically active excitatory relay neurons; RG-F, flexor center, RG-E, extensor center. For target neurons with copies in both, the cervical and the lumbar circuits, connection weights are identical unless otherwise noted.

DOI: https://doi.org/10.7554/eLife.43587.004

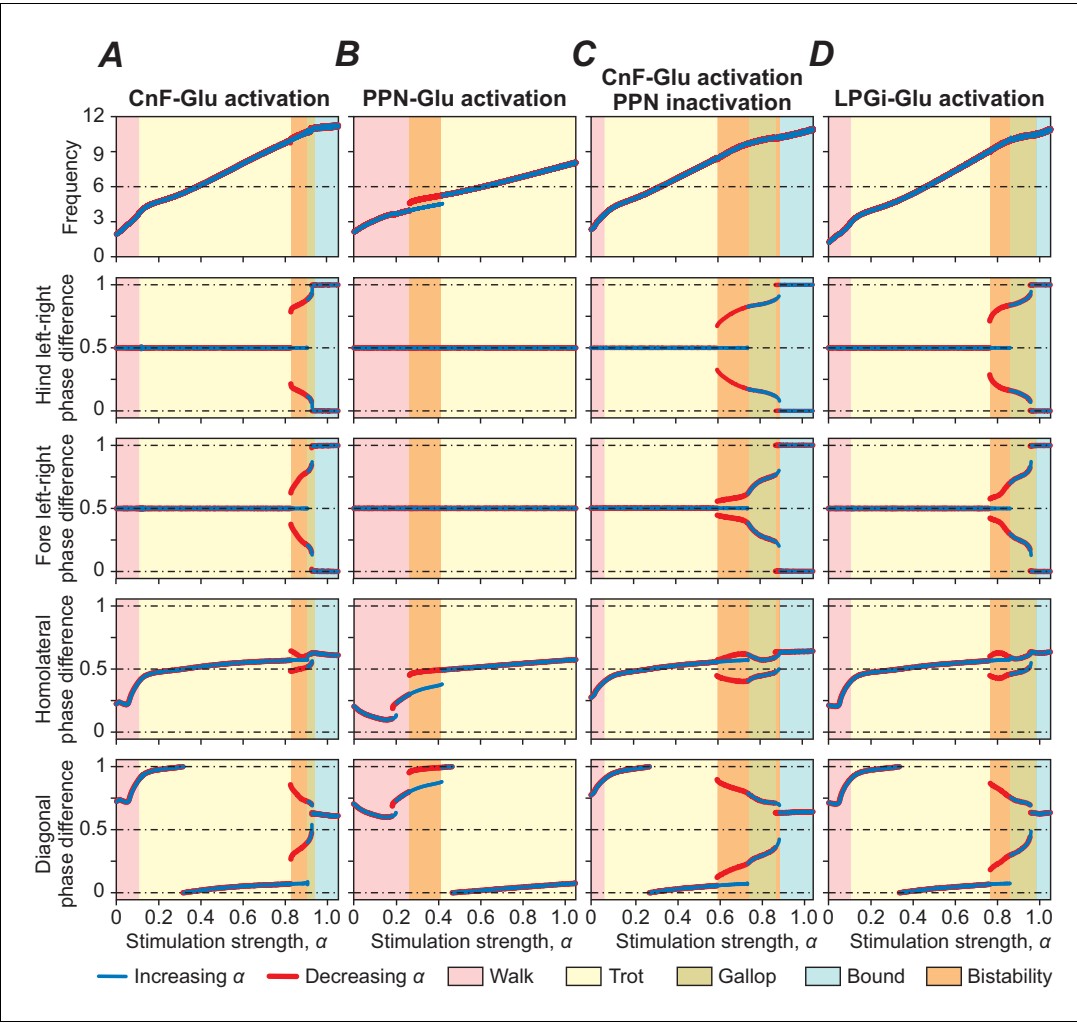

**Figure 3.** Bifurcation diagrams showing the effects of unilateral stimulation of glutamatergic (Glu) populations in the MLR and RF on locomotor frequency (top diagrams) and phase relationships between RGs controlling different limbs, representing different gaits. (**A**) Unilateral stimulation of CnF Glu neurons ($m_{CnF}$ = 1.35; $b_{CnF}$ = 3.95) elicited locomotion with a wide range of frequencies with all four gaits expressed depending on stimulation strength ($\alpha$). The results of these simulations closely correspond to the results of simulations using our previous model (***Danner et al., 2017***). Gait analyses for (**A**) are shown in ***Figure 3—figure supplement 1***. (**B**) Unilateral stimulation of PPN Glu neurons ($m_{PPN}$ = 1.5; $b_{PPN}$ = 4) produced only lower locomotor frequencies and alternating gaits: walk and trot. (**C**) Unilateral activation of CnF Glu neurons ($m_{CnF}$ = 2.55; $b_{CnF}$ = 4.2), while PPN activity was suppressed bilaterally, generated all four gaits but maximum frequency was slightly reduced. (**D**) Unilateral activation of all LPGi Glu neurons ($m_{LPGi}$ = 1.1; $b_{LPGi}$ = 2.45) produced locomotor frequencies and gaits similar to those shown in (**A**). Normalized phase differences of 0.5 correspond to alternation, whereas differences of 0 and 1 correspond to synchronization. Blue and red lines indicate stable phase differences with stepwise increase and decrease of the bifurcation parameter $\alpha$, respectively. Colored areas indicate the expressed gaits or regions of bistability between two adjacent gaits. Bifurcation diagrams are calculated as described in ***Danner et al. (2017)***.

DOI: https://doi.org/10.7554/eLife.43587.005

The following figure supplement is available for figure 3:

**Figure supplement 1.** Illustration of different gaits elicited by unilateral stimulation of glutamatergic neurons in the CnF (shown in ***Figure 3A***).

DOI: https://doi.org/10.7554/eLife.43587.006

*(2016)* and *Danner et al. (2017)*. Since the structure of connections within the cord remained identical to our previous study (*Danner et al., 2017*), both the sequence of the expressed gaits and the intermediate regimes of bistability in *Figure 3A* reproduce our previous results.

2. Unilateral selective activation of glutamatergic neurons in the dorsal PPN in the mouse can only trigger low speed locomotion and generate alternating gaits (walk and trot), even if stimulated at high intensities (*Caggiano et al., 2018*). Our model reproduced this experimental observation since glutamatergic PPN neurons, exciting only the LPGi-Glu-1 populations, had relatively weak synaptic influence on these populations not allowing the generation of high locomotor frequencies (see *Figure 3B*, top diagram). Moreover, the system could not switch from alternating (walk and trot) to synchronous (gallop and bound) gaits, since PPN neurons did not excite the LPGi-Glu-2 populations and, correspondingly, did not activate the transitions to synchronous gaits via the LPGi-Glu-2-spinal cord pathways projecting to spinal CINs and LPNs (see *Figure 3B*).

3. Unilateral selective activation of glutamatergic CnF neurons while PPN activity is suppressed bilaterally in the mouse leads to a decrease in maximum locomotor frequency with all gaits present (*Caggiano et al., 2018*). Corresponding to these experimental results, in our model, unilateral activation of CnF glutamatergic neurons generated locomotor oscillations with all gaits expressed when PPN neurons were inactivated bilaterally but with slightly reduced maximum frequencies (*Figure 3C*). Thus, both CnF and PPN controlled low-frequency alternating-gait locomotion in the model, but only activation of the CnF (with or without the PPN) produced high frequency and left-right synchronous gaits like gallop and bound.

While stimulation of the MLR can initiate and support locomotion of different frequencies and gaits, it does so - in the biological system (*Capelli et al., 2017*) as well as in our model - through activation of glutamatergic neurons in the RF, whose axons descend bilaterally to spinal circuits and presumably activate spinal RGs. Specifically, *Capelli et al. (2017)* have recently shown that unilateral selective stimulation of glutamatergic neurons of the LPGi, a structure within the medullary RF, can initiate locomotion and elicit locomotor oscillations and gaits in a range corresponding to that of selective CnF stimulation. Our model was able to reproduce these findings as well. Unilateral stimulation of glutamatergic LPGi populations in the model produced a similarly wide range of locomotor frequencies and corresponding gaits as was the case for unilateral stimulation of glutamatergic neurons in the CnF (*Figure 3D*). This suggests that the RF, and particularly the LPGi, is involved in mediating MLR control of locomotion.

Together, the above simulations have demonstrated that the proposed brainstem-spinal cord connectome allowed our model to reproduce the experimentally observed effects of stimulation of glutamatergic populations within the CnF, PPN, and LPGi.

## Frequency-dependent gait expression and the effects of PPN inactivation

To explicitly examine the dependence of distinct locomotor gaits on locomotor frequency, the bifurcation diagrams of *Figure 3A and C* were rebuilt to plot changes in phase difference against locomotor frequency (*Figure 4A*). We then compared locomotor gaits when glutamatergic neurons of the left CnF where stimulated (*Figure 4A* top diagram, same simulation as in *Figure 3A*) with the same stimulation while the PPN was bilaterally inactivated (*Figure 4A*, bottom diagram, same simulation as *Figure 3C*). The inactivation of the PPN shifted the transition from alternating gaits (walk and trot) to synchronized gaits (gallop and bound) to lower locomotor frequencies.

The mechanism of this shift in our model is the following. As described above, the LPGi-Glu-1 pathways control locomotor frequency and the LPGi-Glu-2 pathways control gait by promoting the transition from alternating to synchronous gaits (*Figures 1* and *2*). When the PPN was bilaterally inactivated, only the drives to LPGi-Glu-1 were reduced since the PPN on each side only projects to LPGi-Glu-1 while the drive to LPGi-Glu-2 (promoting the transition to synchronous gaits) on each side remained unaffected (*Figures 1* and *2*). Thus, frequency increased more slowly with increasing CnF stimulation and the transition to synchronous gaits (gallop and bound) occurred at lower locomotor frequencies (*Figure 4A,B*). This shift was even more pronounced after increasing noisy current in all neurons ($\sigma_{Noise}$ increased from 0.005 pA to 1.0 pA; *Figure 4C*). Incorporating a moderate noise allowed us to reproduce a natural step-to-step variability and variable frequency-dependent changes similar to those during natural locomotion. Phase differences were evaluated for each step cycle and plotted in equally spaced bins between 0 and 1 over the corresponding locomotor frequency

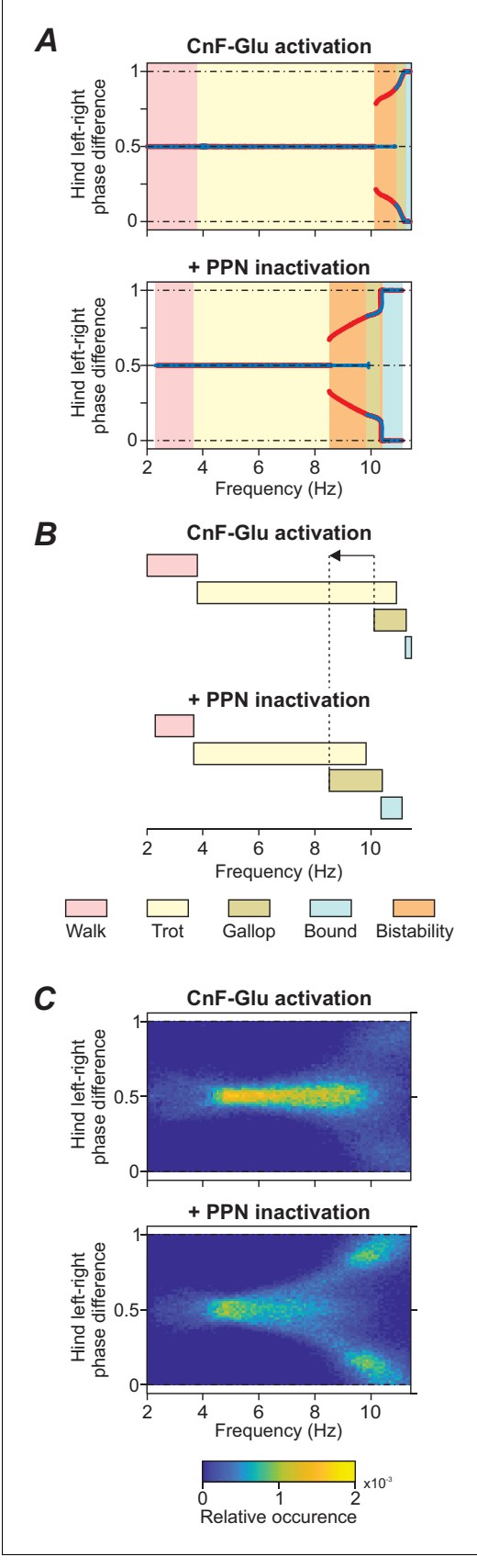

**Figure 4.** Frequency shift in gait transitions following
*Figure 4 continued on next page*

(partitioned in 0.25 Hz bins). All values were then normalized to the total number of bins and can be interpreted as the relative frequency of occurrence of each phase difference-frequency pair (see Materials and methods). *Figure 4C* shows that inactivation of the PPN, while the left CnF glutamatergic population was stimulated, resulted not only in a shift of the transition from alternating to synchronized gaits to a lower frequency (*Figure 4A,B*) but also affected the stability of the steady-state gaits. This can be seen by an increased variability of the left-right phase differences around their stable-state solutions when the PPN is inactivated. The variability increased with increasing frequency and resulted in a wider transition period between trot (left-right alternation) and gallop (left-right synchronization).

## Analysis of relative probabilities of gait expression

Incorporation of step-to-step variability, as described at the end of the previous section, also allows the analysis of variable gait expression as observed in natural locomotion. To approximate frequency-dependent gait expression under noise conditions, we calculated frequency-dependent relative probabilities of expression of each gait (see Materials and methods) for three cases: when stimulation was applied to either CnF, PPN, or LPGi glutamatergic neurons (*Figure 5*). Activation of glutamatergic neurons in the CnF with fixed noise ($\sigma_{Noise}$ = 1 pA) and varying drives ($\alpha$) produced a wide range of frequencies and expression of left-right alternating (walk and trot) as well as left-right synchronous gaits (gallop and bound), while activation of glutamatergic neurons in the PPN resulted in lower maximum frequencies and only expression of left-right alternating gaits (*Caggiano et al., 2018*). While gait distributions were not reported for LPGi stimulations our model suggests that LPGi stimulation can induce both alternating and synchronized gaits (*Figure 5C*). Those gait transitions are expected given the relationship between gait and speed.

## Role of brainstem inhibitory neurons in modulating locomotion

Unilateral activation of inhibitory neurons in the CnF, PPN, or LPGi decelerates or stops ongoing locomotor activity (*Capelli et al., 2017*; *Caggiano et al., 2018*). In freely behaving mice, optogenetic activation of CnF inhibitory neurons (defined by expression of the Vgat promoter) reduces locomotor speed and, in some trials, halts locomotion completely, while activation of

*Figure 4 continued*

PPN inactivation. (**A**)-(**C**) Comparison of model behavior when CnF glutamatergic neurons were stimulated unilaterally without (top graphs) and with (bottom graphs) bilateral inactivation of the PPN. (**A**) Hindlimb left-right phase differences plotted against locomotor frequency using data from *Figure 3A and C*. Comparison of the top and bottom diagram shows that the transition from alternating gaits - walk and trot - to synchronous gaits - gallop and bound - shifted towards lower frequencies when the PPN was bilaterally inactivated. (**B**) Schematic representation of this shift. Dashed lines and arrow indicate the shift of the beginning of gallop. (**C**) Step-by-step variability for hind left-right phase differences illustrates that synchronous gaits were also shifted to lower frequencies on a step-by-step basis. Normalized phase differences of 0.5 correspond to alternation, whereas phase differences of 0 and 1 correspond to synchronization. Blue and red lines in (**A**) indicate the stable phase differences with stepwise increase and decrease of the bifurcation parameter $\alpha$, respectively. Colored areas indicate the expressed gaits or regions of bistability between two adjacent gaits. Bifurcation diagrams are calculated as described in *Danner et al. (2017)*. In (**C**), step-by-step variability with increased noise was calculated as described in Materials and methods.

DOI: https://doi.org/10.7554/eLife.43587.007

PPN inhibitory neurons reduces locomotor frequency to a much lesser extent with only occasional stopping of locomotion (Extended Data Figure 3 in *Caggiano et al., 2018*). The optogenetic activation of LPGi inhibitory neurons, like that of CnF inhibitory neurons, reduces locomotor speed down to a complete halt of locomotion at higher stimulation intensities (Figure 2g in *Capelli et al., 2017*).

These experiments were performed in freely walking mice, thus to simulate the ongoing locomotor activity in the model triggering the whole movement behavior from slow to high speed, we bilaterally activated CnF glutamatergic neurons by applying a constant excitatory tonic drive ($D_{CnF,ex}$) to these neurons. In each series of simulations, the inhibitory neurons in one region of interest (CnF-GABA/Gly, PPN-GABA/Gly, or LPGi-GABA/Gly) were stimulated unilaterally by applying monotonically increasing excitatory drive to the neurons ($D_{CnF/PPN/LPGi,in}$) within the corresponding region. The effects of these stimulations depended on the stimulated region and the initial frequency of the locomotor activity defined by $D_{CnF,ex}$.

Progressive unilateral activation of inhibitory neurons within the CnF (by increasing $D_{CnF,in}$) resulted in a reduction of locomotor frequency and an orderly progression from bound to gallop, trot and walk, and finally stopped locomotion (*Figure 6A* and examples for $D_{CnF,ex}$ = 3.04 in *Figure 6D*).

The same unilateral progressive activation of inhibitory population within the PPN could only decrease locomotor frequency and was not able to stop locomotion (*Figure 6B and E*).

Both these simulations were qualitatively consistent with the experimental data of *Caggiano et al. (2018)*, and systematically demonstrate the possible role of inhibitory neurons in the CnF and PPN in the regulation of locomotor frequency and gait expression.

Unilateral stimulation of the LPGi inhibitory population had a similar effect on locomotor activity to that of stimulation of the CnF inhibitory neurons (*Figure 6C*). Moreover, similar to the experimental studies of *Capelli et al. (2017)* (their Figure 2f,g), progressive activation of glutamatergic populations in the CnF (increase in $D_{CnF,ex}$ at fixed $D_{LPGi,in}$) in our model increased locomotor frequency, whereas progressive activation of the inhibitory population (increase in $D_{LPGi,in}$ at fixed $D_{CnF,ex}$) decreased locomotor frequency up to termination of locomotor oscillations. Also, as is the case of activation of the inhibitory population in the CnF, our model predicts that the decrease of frequency with progressive activation of the inhibitory LPGi population is accompanied by orderly gait transitions (*Figure 6F*).

Importantly, despite similarities of the effects, the underlying mechanisms for frequency reduction with activation of LPGi inhibitory neurons was different from that in the CnF. While activation of inhibitory neurons in the CnF reduced CnF and PPN activity locally within the MLR, LPGi inhibitory neurons suppressed the activity of their downstream targets in the spinal cord, in addition to local glutamatergic neurons in the LPGi (see *Figures 1* and *2*).

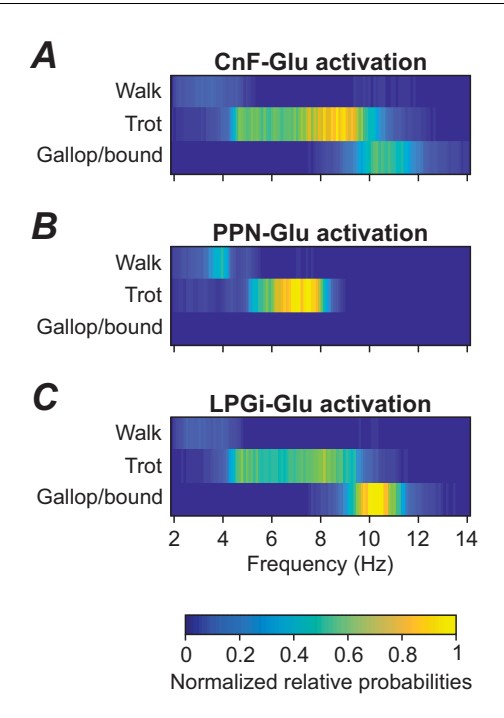

**Figure 5.** Frequency-dependent distribution of gaits caused by unilateral stimulation of glutamatergic neurons in the CnF, PPN, and LPGi. (**A**) Unilateral stimulation of glutamatergic neurons in the CnF resulted in frequency-dependent expression of all gaits: walk, trot and gallop/bound. (**B**) Unilateral stimulation of glutamatergic neurons in the PPN elicited only alternating gaits, walk and trot, at a lower frequency range. (**C**) Unilateral stimulation of glutamatergic neurons in the LPGi resulted in frequency-dependent expression of all gaits similar to that in (**A**). The relative probabilities of frequency-dependent gait expression were analyzed as described in Materials and methods.
DOI: https://doi.org/10.7554/eLife.43587.008

## Discussion

### Brainstem-spinal cord pathways and mechanisms for control of locomotor speed and gait

In our previous modeling studies (*Danner et al., 2016*; *Danner et al., 2017*) we suggested that locomotor speed and gait (limb coordination) are controlled by descending brainstem drives to different targets within the spinal cord. Specifically, locomotor speed (which is dependent on the frequency of the locomotor rhythm) is determined by descending brainstem drives to the locomotor rhythm generating circuits (RGs) controlling the limbs, whereas the phase relationships between these RGs (defining inter-limb coordination and thus gait) is controlled by descending brainstem drives to the specific CIN and LPN populations mediating left-right and fore-hind interactions between the RGs. However, the previous models did not include brainstem centers and the brainstem drives were simply introduced as external inputs. In the present model, we explicitly simulated brainstem circuits and their descending pathways to the spinal cord.

To simulate MLR-controlled locomotion, the brainstem model included two bilaterally located major MLR nuclei (CnF and PPN) whose outputs converged at the left and right LPGi nuclei in the RF, which in turn projected to spinal locomotor circuits (*Liang et al., 2011*; *Liang et al., 2016*; *Capelli et al., 2017*; *Brownstone and Chopek, 2018*; *Caggiano et al., 2018*; *Josset et al., 2018*; *Oueghlani et al., 2018*). The decision to include only the LPGi nucleus of the RF into our model was based on results by *Capelli et al. (2017)*, who had been able to initiate locomotion by optogenetic activation of glutamatergic neurons only in this nucleus of the RF and reported that the LPGi receives projections from both CnF and PPN. However, locomotion could also be evoked via other RF-mediated pathways not involving the LPGi (reviewed in *Brownstone and Chopek, 2018*). Indeed, when *Capelli et al. (2017)* removed LPGi glutamatergic neurons, slow locomotion could still be observed, suggesting that other nuclei can also be involved in mediating MLR pathways to the spinal cord. Hence, what in the model is defined as LPGi may also include other subpopulations of glutamatergic neurons within or even outside the RF. The identity and location of these neurons are not known at this moment.

Finally, based on our simulations we suggest that each (left and right) LPGi has separate glutamatergic populations that give rise to two separate pathways controlling locomotor frequency through activation of spinal rhythm generating circuits and gait via regulation of specific spinal CIN and LPN populations (*Figures 1* and *2*). We also suggest that the activities of these LPGi populations are mediated and distributed within the spinal cord bilaterally and between cervical and lumbar circuits by ipsi- and contralaterally projecting populations of interneurons (such as the Ine, CINe, and dIni populations in *Figure 2*). Such spinal interneurons that receive descending inputs from the RF and distribute their activity widely within the spinal cord have been found in cats (*Jankowska et al., 2003*; *Matsuyama et al., 2004*) and rats (*Mitchell et al., 2016*). These suggestions await experimental testing in the future.

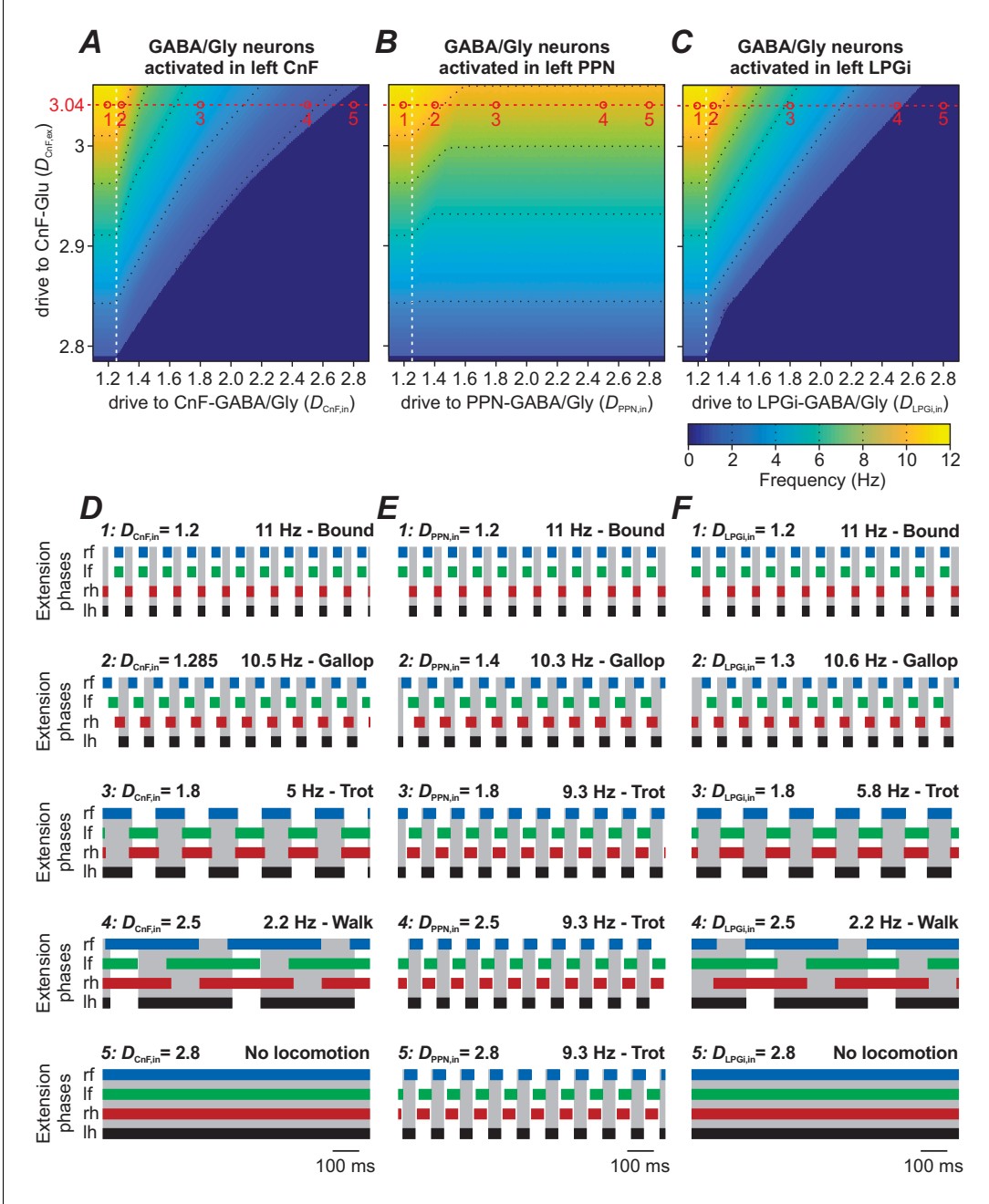

**Figure 6.** Role of inhibitory neurons within the CnF, PPN and LPGi in modulating locomotion. In all simulations, locomotor oscillations were produced by bilateral activation of glutamatergic populations in the CnF by the variable excitatory drive, $D_{CnF,ex}$. The other variable drive was applied unilaterally to the inhibitory population either within the CnF ($D_{CnF,in}$, **A** and **D**) or within the PPN ($D_{PPN,in}$, **B** and **E**), or within the LPGi ($D_{LPGi,in}$, **C** and **F**). In (**A**), (**B**) and (**C**), the corresponding 2D diagrams were built for all three cases, and frequency was represented by color. (**A**) Unilateral stimulation of the inhibitory population in the CnF reduced locomotor frequency and stopped locomotion at higher stimulation intensities. (**B**) Unilateral stimulation of the inhibitory population in the PPN decreased locomotor frequency but was not able to arrest locomotor oscillations completely. (**C**) Unilateral stimulation of the inhibitory population in the LPGi decreased locomotor frequency and could also stop locomotion similar to the situation in (**A**). Black dotted lines indicate iso-frequency lines for 2, 4, 6, 8 and 10 Hz. White vertical dashed lines indicate the threshold for activation of the corresponding inhibitory populations. (**D**)-(**F**) Example traces of rhythmic extensor activities in all four RGs to illustrate changes in gait for the different stimulation parameters. An increase of inhibition in all cases was accompanied by sequential frequency-dependent

*Figure 6 continued on next page*

*Figure 6 continued*

gait transitions. Examples 1–5 in (**D**)-(**F**) are taken from the parameter combinations indicated by open circles (labeled 1–5) along the red dashed lines in (**A**)-(**C**). In all examples, $D_{CnF,ex}$ = 3.04, $D_{CnF/PPN/LPGi,in}$ are indicated for each simulation. lh: left hind; lf: left fore; rh: right hind; rf: right fore.
DOI: https://doi.org/10.7554/eLife.43587.009

## Distinct brainstem-spinal cord pathways for control of slow and fast locomotion

Recent studies in mice raised a possibility that slow exploratory-type locomotion and fast escape-like locomotion might be initiated and controlled by distinct brainstem circuits. Specifically, it has been shown that slow locomotion can be evoked by activation of glutamatergic neurons in the PPN (*Caggiano et al., 2018*), whereas fast locomotion is initiated and controlled by glutamatergic neurons in the CnF (*Caggiano et al., 2018*; *Josset et al., 2018*). The role of the PPN in the generation (*Caggiano et al., 2018*) and control (*Josset et al., 2018*) of slow locomotion appears to be more complex. In contrast to *Caggiano et al. (2018)*, *Josset et al. (2018)* conclude that activation of PPN glutamatergic neurons in resting animals can only activate flexor muscles but is not very effective in initiating locomotion per se. Yet, similar to *Caggiano et al. (2018)*, the suppression of the PPN in *Josset et al. (2018)* also reduces the locomotor speed.

To simulate and propose a mechanistic explanation for the functional difference between the activation of the PPN and CnF we incorporated bilaterally interacting CnF and PPN nuclei (*Figures 1* and *2*). In our model, in addition to ipsi- and contralateral projections from the glutamatergic CnF populations to the glutamatergic PPN populations, both CnF and PPN glutamatergic populations project bilaterally to the left and right LPGi nuclei. However, we suggest that the projections from the PPN target only LPGi-Glu-1 populations (responsible for control of locomotor frequency, but not gait), whereas the projections from the CnF affect both LPGi populations (LPGi-Glu-1 and LPGi-Glu-2) and hence can control locomotor frequency and provide frequency-dependent control of locomotor gait (*Figures 1* and *2*). In addition, we assume that synaptic weights of excitatory projections from the CnF to LPGi-Glu-1 are much stronger than those from the PPN. Therefore, in our model, a unilateral progressive activation of the glutamatergic population within the CnF results in monotonic increase of locomotor frequency up to 10–11 Hz accompanied by frequency-dependent gait transitions from walk to trot to gallop and bound (*Figures 3A* and *5A*). These results are consistent with experimental studies of *Caggiano et al. (2018)* (their Figure 1h–j) and *Josset et al. (2018)* (their Figure 6c1), providing indirect validation of the network architecture implemented in our model. In contrast, a unilateral progressive activation of the glutamatergic population within the PPN can only produce an increase of frequency up to ~8 Hz accompanied by only left-right asymmetric gaits, such as walk and trot (*Figures 3B* and *5B*). These results are in agreement with experimental studies of *Caggiano et al. (2018)* (their Figure 1h–j) and support the observations of *Josset et al. (2018)* (their Figure 6c2 and e4) of a maintenance of asymmetrical walking at subthreshold or threshold PPN activations.

A unilateral progressive stimulation of glutamatergic populations in the LPGi produces effects - similar to those during unilateral monotonic activation of the glutamatergic population in the CnF - a monotonic increase of locomotor frequency and frequency-dependent gait transitions (*Figures 3D* and *5C*). While the increase of frequency during unilateral monotonic activation of glutamatergic neurons in the LPGi is consistent with the results of *Capelli et al. (2017)*, the authors of this study did not analyze gaits during their experiments, but our results suggest sequential gait changes with increasing frequencies similar to that of CnF stimulation.

## The effects of PPN inactivation on the locomotor speed and gaits

Inactivation of glutamatergic neurons in the PPN leads to an overall reduction of locomotor speed (*Caggiano et al., 2018*; *Josset et al., 2018*). More specifically, *Caggiano et al. (2018)* studied the effect of pharmacological inactivation of the PPN on locomotor characteristics. They found that inactivation of the PPN led to a reduction of locomotor speed, but the animals were still able to express the full spectrum of locomotor gaits. Our simulations were qualitatively consistent with this conclusion (see *Figure 3C*). Moreover, our simulations have shown that the suppression of the PPN shifts

the transition from alternating to synchronized gaits towards lower locomotor frequencies (*Figure 4A,B*). This shift is even more pronounced after adding noise to the model (*Figure 4C*).

### Role of brainstem inhibitory neurons in modulating locomotion

It has been shown that unilateral optogenetic stimulation of Vgat inhibitory neurons within the CnF in freely moving mice reduces locomotor speed and can fully stop locomotion, while the same stimulation of inhibitory neurons in the PPN affects locomotor speed to a lesser extent and usually does not stop locomotion (*Caggiano et al., 2018*, their Extended Data Figure 3a–c). The fact that optogenetic activation of inhibitory neurons in the CnF can fully suppress locomotor activity suggests that CnF inhibitory neurons inhibit the PPN, which was implemented in our model (*Figures 1* and *2*). These inhibitory connections, however have not been shown experimentally. Optogenetic activation of inhibitory neurons in the RF including the LPGi progressively decreases locomotor speed and can fully stop locomotion (*Capelli et al., 2017*, their Figure 2f,g and Extended Data Figure 6d,h).

Our simulations were qualitatively consistent with these experimental findings (*Figure 6*). Notably, in our simulations, the decrease of locomotor frequency during unilateral activation of inhibitory populations in the CnF, PPN and LPGi was always accompanied by orderly progressive changes of locomotor gaits toward slower gaits like trot and walk (*Figure 6D–F*). Although *Caggiano et al. (2018)* and *Capelli et al. (2017)* did not analyze gait transitions during progressive increase of stimulation intensity to inhibitory neurons in these areas, our findings suggest that frequency/gait relations are not affected by unilateral activation of inhibitory neurons in these areas.

### Model limitations

In this study we aimed to reproduce new experimental data on the brainstem control of locomotion and the specific role of CnF and PPN nuclei and the RF (*Capelli et al., 2017*; *Caggiano et al., 2018*; *Josset et al., 2018*) within the framework of our previous model (*Danner et al., 2017*). Correspondingly, we incorporated MLR and RF circuits into the model with only minimal changes to the original network structure. This ensures that our previous model assumptions and conclusions hold true for the extended model as well, but at the same time increases the complexity of the model to a point where thorough mathematical and performance analyses are not feasible. Therefore, in future studies it will be beneficial, in parallel with detail models, to generate related, but simplified network models allowing systems level mathematical analysis and more thorough investigation of the dynamic mechanisms underlying the brainstem control of locomotor frequency and gait. An example of such simplifications and systems level analyses has been performed by others for our previous model (*Lodi et al., 2017*; *Lodi et al., 2018*),

Similar to our previous modeling investigations (*Danner et al., 2016*; *Danner et al., 2017*), in this study we focused only on central neural interactions, without considering other brainstem circuits beyond MLR and RF and circuits in the spinal cord operating below the RGs, such as reflex circuits and motoneurons. Neither have we developed a full biomechanical model to investigate the role of biomechanics and sensory feedback in the control of locomotor speed and gait. We also did not take into account heterogeneous activity profiles of individual neurons (*Caggiano et al., 2018*; *Oueghlani et al., 2018*) as populations are represented by an activity based model. All the above will be the focus of future investigations.

## Materials and methods

### Model architecture

The model represents a bilateral network of interconnected populations of neurons and includes the simulated brainstem and spinal cord compartments. The present model was based on, and represents an extension of, our previous model (*Danner et al., 2017*). While keeping the same spinal cord circuitry we added brainstem compartments including bilaterally interacting CnF and PPN compartments as well as LPGi compartments mediating descending brainstem drives to the spinal cord (*Figures 1* and *2*). Several additional relay neuron populations (Ine, CINe, and dIni) were incorporated to mediate and distribute brainstem signals to the spinal cord bilaterally and between the cervical and lumbar compartments of the cord (*Figure 2*, more detailed description of new network structures in Results).

## Models of neuron populations and model parameters

Each population in the model was represented by a non-spiking, 'activity-based' model (*Ermentrout, 1994*). The flexor and extensor RG centers (F and E populations, see *Figure 2*) incorporated a persistent sodium current and had intrinsic oscillating properties. The average membrane potential, $V$, in these populations was described as:

$$C \cdot dV/dt = -I_{\text{NaP}} - I_{\text{L}} - I_{\text{SynE}} - I_{\text{SynI}} - I_{\text{Noise}}. \tag{1}$$

In all other populations, the average membrane potential obeyed the following equation:

$$C \cdot dV/dt = -I_{\text{L}} - I_{\text{SynE}} - I_{\text{SynI}} - I_{\text{Noise}}, \tag{2}$$

where $C$ is the membrane capacitance, $I_{\text{NaP}}$ the persistent sodium current, $I_{\text{L}}$ the leak current, $I_{\text{SynE}}$ and $I_{\text{SynI}}$ excitatory and inhibitory synaptic currents, respectively, and $I_{\text{Noise}}$ a noisy current. The output function $f(V)$ translates $V$ into the integrated population activity representing population output as defined by the linear piecewise function:

$$f(V) = \begin{cases} 0, & \text{if } V < V_{\text{thr}} \\ (V - V_{\text{thr}})/(V_{\text{max}} - V_{\text{thr}}), & \text{if } V_{\text{thr}} \leq V < V_{\text{max}} \\ 1, & \text{if } V \geq V_{\text{max}} \end{cases} \tag{3}$$

A complete description of the population model and parameters are presented in *Danner et al. (2017)*. In all brainstem (CnF, PPN, and LPGi) and relay (Ine, CINe, and Ini) neuron populations the conductance variable, $g_{\text{L}}$, was equal to 5 nS.

Synaptic connection weights were adapted from our previous model (*Danner et al., 2017*) and weights for newly introduced connections were selected within their operating ranges and tuned to produce gait transitions similar to those of our model from *Danner et al. (2017)* and to reproduce the relevant experimental data (*Capelli et al., 2017*; *Caggiano et al., 2018*). Connection weights are listed in *Table 1*.

To simulate the effect of activation of a neuron population i (i ∈ [CnF, PPN, LPGi, VN]) in the brainstem, we applied a tonic excitatory drive, $D_{\text{i,j}}$ to this population. The stimulation strength $D_{\text{i,j}}$ was given by the following equation:

$$D_{\text{i,j}}(\alpha) = m_{\text{i}} \cdot \alpha + b_{\text{i}}, \tag{4}$$

where $m_{\text{i}}$ is the slope and $b_{\text{i}}$ the intercept. The scalar $\alpha \in [0, 1.05]$ characterizes the variable stimulation strength. The index j (j ∈ [ex, in]) indicates if excitatory (ex, Glu) or inhibitory (in, GABA/Gly) populations are stimulated.

## Computer simulations and data analysis

The set of differential equations was solved with the same custom C++ code using odeint of the boost library used in *Danner et al. (2017)*. The C++ code was compiled as a python module and python 3.6 was used to interface with the simulation and to analyze the results. Source code and python scripts to create all simulations presented here are available on GitHub at https://github.com/SimonDanner/CPGNetworkSimulator (*Danner, 2019*; copy archived at https://github.com/elifesciences-publications/CPGNetworkSimulator). Data analysis procedures are described in *Danner et al. (2017)*.

## Analysis of model performance

Similar to our previous models (*Shevtsova et al., 2014*; *Shevtsova et al., 2015*; *Danner et al., 2016*; *Danner et al., 2017*), the extensor RG centers in the current model were in a tonic mode while the flexor RG centers were oscillating.

To produce locomotor activity by activation of glutamatergic populations within the CnF, PPN, or LPGi, the excitatory drive was unilaterally applied to the population of interest. For each of these simulations, values for $m_{\text{i}}$ and $b_{\text{i}}$ (*Equation 4*) are indicated in the corresponding figure legends. To simulate inactivation of the PPN region, all weights of connections originating from this region were set to 0.

The bifurcation diagrams (*Figure 3*) were built for four normalized phase differences [hind left-right, fore left-right, homolateral (left fore – left hind) and diagonal (right fore – left hind)]. To this end, $\alpha$ was increased from 0.0 to 1.05 and then decreased back to 0.0 in 1000 equally spaced steps (for details see *Danner et al., 2017*). At each step, simulations were performed in 10 s intervals until the standard deviation of each phase-difference measured over five locomotor cycles was less than 0.001 or 200 s passed. The frequency of oscillations as the reciprocal of the period was calculated and the hind left-right bifurcation and phase-transition diagrams were built versus frequency to compare model performance in the cases when the CnF was activated with and without PPN inactivation (*Figure 4*).

To consider step-by-step variability (*Figure 4C*) and relative probabilities of frequency-dependent gait expression (*Figure 5*), simulations were performed with increased noisy currents ($\sigma_{Noise}$ = 1 pA, see Equation (14) in *Danner et al., 2017*). To this end, the free parameter $\alpha$ was increased from 0.0 to 1.05 in steps of 0.01. At each step, the simulation was run for 100 s. For the left-right hindlimb phase difference and for the gait, bivariate histograms were created with cycle frequency as the second variable. The phase difference was partitioned into 65 equally spaced bins between 0 and 1 and the frequency was partitioned into 0.25 Hz wide bins from 0 to 14 Hz (*Figure 4C*). Gaits were evaluated at each step cycle based on the definition in Table 2 of *Danner et al. (2017)* and gallop and bound were grouped together (*Figure 5*). The counts per 2D-bin were then divided by the total number of locomotor cycles. Thus, these numbers represent the relative frequency of occurrence of each phase difference-frequency or gait-frequency pair and can be interpreted as a probability.

To simulate the effect of activation of inhibitory (GABA/Gly) neurons in the CnF, PPN, or LPGi, the locomotor-like activity was initially evoked by bilateral application of excitatory drive to glutamatergic neurons in the CnF ($D_{CnF,ex} \in$ [2.78, 3.06] in 0.02 steps). Then, to simulate activation of inhibitory (Gly/GABA) populations in the CnF, PPN, or LPGi, for each value of $D_{CnF,ex}$, the excitatory drive was unilaterally applied to the corresponding inhibitory population ($D_{CnF/PPN/LPGi,in} \in$ [1.15, 2.85] in 0.07 steps).

To test the robustness of the model, we simultaneously varied all connection weights by multiplying each weight by a normally distributed random number with a mean of 1 and standard deviation $\sigma_p$ between 0.02 and 0.2 in steps of 0.02. For each $\sigma_p$, 100 random models were built and bifurcation diagrams were calculated. With $\sigma_p \leq$ 0.04 all randomized models retained all stable regimes and their sequential transitions with changes of $\alpha$. With increasing $\sigma_p$ an increasing number of models lost some stable solutions (gaits such as bound or trot) and 50% of the models were unstable at $\sigma_p$ = 0.2. Thus, the final model represents a coarse system allowing parameter variations without dramatic (qualitative) changes in behavior.

# Additional information

## Funding

| Funder | Grant reference number | Author |
|---|---|---|
| National Institutes of Health | R01NS095366 | Natalia A Shevtsova |
| National Institutes of Health | R01NS090919 | Ilya A Rybak |
| College of Medicine, Drexel University | Edward Jekkal Muscular Dystrophy Association Fellowship | Jessica Ausborn |

The funders had no role in study design, data collection and interpretation, or the decision to submit the work for publication.

## Author contributions

Jessica Ausborn, Natalia A Shevtsova, Conceptualization, Software, Formal analysis, Investigation, Visualization, Writing—original draft, Writing—review and editing; Vittorio Caggiano, Conceptualization, Validation, Writing—review and editing; Simon M Danner, Conceptualization, Software, Formal analysis, Visualization, Writing—review and editing; Ilya A Rybak, Conceptualization, Supervision, Validation, Writing—original draft, Writing—review and editing

## Author ORCIDs

Jessica Ausborn ![ORCID] http://orcid.org/0000-0003-4500-5131
Natalia A Shevtsova ![ORCID] http://orcid.org/0000-0002-1971-9707
Vittorio Caggiano ![ORCID] http://orcid.org/0000-0002-2186-1550
Simon M Danner ![ORCID] https://orcid.org/0000-0002-4642-7064
Ilya A Rybak ![ORCID] https://orcid.org/0000-0003-3461-349X

## Decision letter and Author response

Decision letter https://doi.org/10.7554/eLife.43587.013
Author response https://doi.org/10.7554/eLife.43587.014

## Additional files

### Supplementary files

• Transparent reporting form
DOI: https://doi.org/10.7554/eLife.43587.010

### Data availability

Source code and python scripts to create all simulations presented here are available on GitHub at https://github.com/SimonDanner/CPGNetworkSimulator (copy archived at https://github.com/elifes-ciences-publications/CPGNetworkSimulator).

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
