## [Decision Letter]

Thank you for submitting your article "Computational modeling of brainstem circuits controlling locomotor frequency and gait" for consideration by *eLife*. Your article has been reviewed by three peer reviewers, including Ronald L Calabrese as the Reviewing Editor and Reviewer #1. The following individuals involved in review of your submission have also agreed to reveal their identity: Robert M Brownstone (Reviewer #2); Maxim Bazhenov (Reviewer #3).

The reviewers have discussed the reviews with one another and the Reviewing Editor has drafted this decision to help you prepare a revised submission.

Summary:

In this manuscript advance, the authors extend their previous model of the spinal circuitry underlying locomotor rhythm generation, and speed and gait control in mice that is based on physiological, behavioral, and genetic studies of identified neuron classes in spinal cord. Here they propose a connectome of the brainstem-spinal circuitry and suggest a mechanistic explanation of the operation of brainstem structures (CnF and PPN of the MLR and downstream LPGi of the caudal RF) and their roles in controlling speed and gait. Their model predicts that brainstem control of locomotion is mediated by two pathways, one controlling locomotor speed via connections to rhythm generating circuits in the spinal cord and the other providing gait control by targeting commissural and long propriospinal interneurons. Specifically they predict that the projections from the PPN target only LPGi-Glu-1 populations (responsible for control of locomotor frequency, but not gait), whereas the projections from the CnF affect both LPGi populations (LPGi-Glu-1 and LPGi-Glu-2) and hence can control locomotor speed and provide speed-dependent control of locomotor gait. This advance should arouse interest in the community studying the neural control of locomotion.

Essential revisions:

1) With respect to the CnF/PPN: the authors are basing their studies mainly on the paper out of the Kiehn lab, while they mention the work from the Bretzner lab. I think the strength of the computational work is the strong support it has from the 2 labs, and the authors could stress this throughout – from the Introduction through the Discussion. That is, there is strong evidence about the roles of these nuclei (the study of which go back decades in many labs). In contrast, the data supporting the LPGi are not as strong, and the authors have relied on a single study that has minimal support, with most previous studies pointing towards the GRN proper as being critical for locomotion (eg Noga, Mori labs). In my view, the specific location(s) of reticulospinal neurons for locomotion thus remains an open question. Having said that, I don't think this detracts from the computational study which does not rely on the specific locations of the neurons. In fact, this study predicts 2 different types of descending systems – must they both be in LPGi? Perhaps one reason for the discrepancies in the literature is that both LPGi and GRN are involved and are home to these different descending families of neurons? I think it could be useful if the authors explored this. (Also, statements like "pivotal role of LPGi" cannot be supported by this study for the same reason.)

2) Where is the dIni population located, and is there evidence to support this prediction?

3) Section on PPN inactivation, second paragraph of subsection “Frequency-dependent gait expression and the effects of PPN inactivation” – I found it very difficult to follow the logic here – which centres on what happens if you don't activate the Glu-1 population. Perhaps it would be clearer to state what happens when you do activate it? I am having trouble relating this to gait transition.

4) As the known connectome of the spinal cord and brainstem expands, one wonders if this process of expanding the model can continue without the models becoming so complicated and contain so many unknown connection weights that they are no longer heuristic. This unfortunate side effect of the process of model building for complicated networks has not overwhelmed the message and insights of this paper but future models building on this one may suffer that fate. Can the authors discuss this potential limitation of their approach?

---

## [Author Response]

Essential revisions:1) With respect to the CnF/PPN: the authors are basing their studies mainly on the paper out of the Kiehn lab, while they mention the work from the Bretzner lab. I think the strength of the computational work is the strong support it has from the 2 labs, and the authors could stress this throughout – from the Introduction through the Discussion.

Our model agrees with most of the experimental results of both Caggiano et al., 2018 and Josset et al., 2018. We have included a number of additional references to the corresponding results of both studies throughout the manuscript and also a brief discussion of their differences in the description of the effects of stimulation of glutamatergic neurons in the PPN (see section "Distinct brainstem-spinal cord pathways for control of slow and fast locomotion" in the Discussion).

That is, there is strong evidence about the roles of these nuclei (the study of which go back decades in many labs). In contrast, the data supporting the LPGi are not as strong, and the authors have relied on a single study that has minimal support, with most previous studies pointing towards the GRN proper as being critical for locomotion (eg Noga, Mori labs). In my view, the specific location(s) of reticulospinal neurons for locomotion thus remains an open question. Having said that, I don't think this detracts from the computational study which does not rely on the specific locations of the neurons. In fact, this study predicts 2 different types of descending systems – must they both be in LPGi? Perhaps one reason for the discrepancies in the literature is that both LPGi and GRN are involved and are home to these different descending families of neurons? I think it could be useful if the authors explored this. (Also, statements like "pivotal role of LPGi" cannot be supported by this study for the same reason.)

We agree with the reviewers that the LPGi is not the only RF nucleus mediating MLR pathways to the spinal cord controlling locomotor speed and gait and that other parts of the RF probably contribute to these functions. Related to this, we would like to note that most of the previous studies of the role of the pontomedullary RF were performed in cats (Noga, Mori, Drew labs), and the existing differences in the terminology, location and maybe even function between the animals create additional difficulties in combining the data. This issue will surely need to be addressed in future studies. To better reflect our (and the reviewers’) view on this issue we have revised the text in several places to clarify the potential role of other RF nuclei. The statement on "the pivotal role of LPGi…" has been replaced with "the RF, and particularly the LPGi, is involved in mediating MLR control of locomotion".

Also, we now discuss this issue in the first section of the Discussion ("Brainstem-spinal cord pathways and mechanisms for control of locomotor speed and gait"). We state there that "what in the model is defined as LPGi may also include other subpopulations of glutamatergic neurons within or even outside the RF. The identity and location of these neurons are not known at this moment". Moreover, as the reviewers mentioned, the results of our modeling study are obviously independent of the exact location of RF nuclei mediating MLR control of locomotion.

2) Where is the dIni population located, and is there evidence to support this prediction?

In Danner et al., 2017, we suggested that gait changes with increasing locomotor speed depend on progressive inhibition of V0 CINs. To achieve this, each glutamatergic descending pathway controlling gait must contain an inhibitory population to mediate inhibition to these CINs (which we named dIni in our model). In the model we suggest that these dIni are located in the spinal cord and inhibit both cervical and lumbar V0 CINs. However, the same could be achieved if the descending glutamatergic neurons activated separate populations of inhibitory interneurons which in turn independently inhibited cervical and lumbar V0 interneurons. Another possibility would be that these inhibitory dInis were located in the brainstem and then project down to cervical and lumbar spinal V0s. To our knowledge there is no direct experimental evidence for the location of such interneurons. However, multiple spinal interneurons that receive descending inputs from the reticular formation and distribute their activity widely within the spinal cord have been found in cats (Jankowska et al., 2003; Matsuyama et al., 2004) and rats (Mitchell et al., 2016). This is mentioned in the manuscript in the last few lines of the first section of Discussion (Brainstem-spinal cord pathways and mechanisms for control of locomotor speed and gait).

3) Section on PPN inactivation, second paragraph of subsection “Frequency-dependent gait expression and the effects of PPN inactivation” – I found it very difficult to follow the logic here – which centres on what happens if you don't activate the Glu-1 population. Perhaps it would be clearer to state what happens when you do activate it? I am having trouble relating this to gait transition.

We apologize and rewrote this section using a bottom up instead of a top down logic. We hope this will make our explanations clearer.

4) As the known connectome of the spinal cord and brainstem expands, one wonders if this process of expanding the model can continue without the models becoming so complicated and contain so many unknown connection weights that they are no longer heuristic. This unfortunate side effect of the process of model building for complicated networks has not overwhelmed the message and insights of this paper but future models building on this one may suffer that fate. Can the authors discuss this potential limitation of their approach?

With the present model, we aimed to reproduce new experimental data with minimal changes to our previous model to ensure that the previous conclusions hold true for the extended model as well and to explore the new data within the framework of our previous predictions. Not surprisingly this increased the complexity of the model. However, we agree with the reviewer about the limitations of this approach when it comes to more thorough analyses of network behavior and underlying mechanisms. It will certainly be beneficial to generate more abstract models in the future. For example, our previous model has already been simplified to allow systems-level analysis of this model and some general conclusions have been made (Lodi et al., 2017, 2018). We have added a paragraph in the Discussion, section “Model limitations”.